# A Business Creation in Post-Industrial Tourism Objects: Case of the Industrial Monuments Route

**Adam R. Szromek** * and **Krzysztof Herman**

Department of Organization and Management, Institute of Economy and Informatics, Silesian University of Technology, Akademicka 2A, 44-100 Gliwice, Poland; krzysztof.herman@polsl.pl

*   Correspondence: innowator@o2.pl; Tel.: +48-32-277-7336

**Abstract:** The aim of this article is to discuss the basic types of business transformations identified in post-industrial heritage sites in the context of changes in business models. The basis for this analysis is the research carried out in 2017 in 42 post-industrial tourism objects, in the *Industrial Monuments Route* (IMR) largest in Poland, that is a part of *European Route of Industrial Heritage*. The analysis of historical changes and the documentation of objects, within the Industrial Monuments Route, made it possible to identify three transformation types in business models of these objects. The post-production organization model can be considered the most popular scheme on the analyzed route. It concerns an enterprise or cultural institution, that previously was a production or extraction plant and currently services tourists only. Although these objects were not designed with tourists in mind, they perfectly fulfill this function due to the presented transformations.

**Keywords:** post-industrial tourism; The Industrial Monuments Route; business models

## 1. Introduction

The concept of post-industrial society, introduced half a century ago by A. Touraine [1], and later popularized by D. Bell [2], is characteristic of the global economy development stage present from the second half of 20th century, where the service sector generates more wealth than the production sector, services employ more employees than production sites, and mass production is replaced by individual production. Knowledge becomes a resource. Therefore, the source of income for societies is the process of production and processing of information, not material goods production. The factor for economy development is the creation of new ideas.

Currently, traces of the post-industrial era can be observed in many economy sectors around the world. The dynamic technological progress at the turn of the millennia accelerated the post-industrialization process. It is particularly evident in the development of IT services that act as the source of almost unlimited customized and global collective communication. The development of new technologies, defined in the post-industrial era as third technological wave [3] (after agricultural and industrial ones), is evident not only in retrospect, as previously, but almost immediately. Without much effort, one can observe the changes referred to above in the global economy. There are, however, also negative consequences of the third wave visible—for example—in the post-industrial landscape, full of abandoned factories with outdated production lines, full of exploited drifts (carbon, silver, gold), that quite often lower the security level of community living in their vicinity and deform the city landscape.

The characteristic feature of the post-industrial era is also the change of technological processes that no longer manufacture material goods but deal in services (non-material goods). It is worth noting that in this case, post-industrial tourism more often becomes an additional activity carried out by industrial plants (for example, multi-generational breweries that, apart from production offer

sightseeing tours, concern both the modern production lines and old beer production technology lines from past centuries). One can take Żywiec brewery as an example, it's one the largest enterprises of this type in Poland. It can also be observed that, sometimes these additional service activities, carried out by industrial enterprises, end in the domination of these tourist services over production. In such cases, the industrial plant becomes a post-industrial tourism object, making regional post-industrial heritage available, similarly to, for example, closed silver or hard coal mines, where the extraction was discontinued, due to economic reasons or due to the lack of deposits. In this aspect, tourist activities become an attempt to make use of non-material (history, ideas) and material values (abandoned machinery and facilities) for tourist purposes, thus contributing to the sustainable development of society. Researchers like, Jonsen-Verbeke, M. [4] have shown industrial heritage as an idea of new tourist product (new value of old things). N. Yashalova et al. [5] has indicated that industrial tourism encourages regional economy growth by stimulating the activities of related sectors.

The article discusses three basic types of business transformations, identified in post-industrial heritage sites, in the context of changes in business models. The basis for this analysis is the research carried out in 2017, in 42 post-industrial tourism objects, associated in the larges part in Poland Industrial Monuments Route (IMR), that is a part of *European Route of Industrial Heritage*. We believe that this article will help understand how the value for customers and other elements of business could be created. Moreover, the article shows how the resources, on which the enterprise was built, affect to other aspects of the business.

## 2. Cultural Heritage and Post-Industrial Tourism

The literature indicates that heritage is what we inherit from the past, use today, and pass on to future generations [6]. Potential heritage resources are vast and widespread, and they include many objects, places, events, persons, and phenomena not heretofore considered to be traditional heritage tourism products [7]. D.J. Timothy indicates that "people are becoming more sophisticated in their travel tastes and desires; many are showing more interest in the deeper meanings of places, local identities, and their own connections to the places they visit".

Human heritage is strongly linked to the culture understood as humankind's spiritual and material heritage [8]. It can also be an element of national identity [9]. It is strongly related to tourist activities. Traveling to visit cultural sites is called cultural tourism, and is one of the fastest developing tourism forms [10].

A. Mikos von Rohrschedit [6] describes cultural tourism as travel during which its participants encounter objects or values, being the manifestation of high or popular culture or expansion of their knowledge about the world surrounding them. Similar concepts on cultural tourism are shared by other researchers. W. Pannich et al. [11] indicate the particular will to learn, discover life styles, art, architecture, and other aspects linked to human existence. Cultural tourism is characterized by intense engagement of human perception. The concept of experience tourism was identified, among others, by C. Hall and H. Zeppel [12]. Others [13] define it as a movement of people to cultural attractions, located outside their place of residence, to gather new information and experiences to satisfy their cultural needs.

Whereas heritage, being an integral part of culture [9], is defined as a mission that is about taking care of historical heritage and maintaining its authenticity as much as possible [14]. H. Park [9] considers it an important element of national history that reminds their citizens about their roots on which the sense of belonging is based.

Undoubtedly, this is a phenomenon characterized by a high sensitivity. J. J. Zhang [15] claims that conflicts may include ethnic clashes, religious differences, or political rivalry, and these difficult pasts are inevitably encountered by tourists from both sides. If such heritage increases the complexities associated with tourism politics, post-conflict tourism becomes even more sensitive when it is associated with historical sites of bloodshed.

According to G. Ismagilov et al. [16], historical and cultural heritage plays a major role when it comes to developing tourism in a given country. When describing the Russian market, they indicated that heritage is a real opportunity to improve the economic, social, and cultural status, thus creating the places that demonstrate that heritage is a sign of creativity and cultural promotion of the local community. Therefore, heritage is more linked with social and economic development [17].

In the context of geography, heritage was discussed by M. Ursache [17], who described cultural heritage as a significant determinant of attractiveness and competitiveness of a given country as a tourist attraction. Due to economic reasons, enterprises focused on cultural heritage should strive to achieve financial independence. It is also emphasized by M. and C. Surugiu [18], in the context of providing support for the entrepreneurship related to it. One must notice, however, that heritage is an economic opportunity for the development of a given area, but on the other hand it maintains social identity. Some claim that it should be protected, not due to financial aspects, but due to its unique value [19].

Due to the above, traveling to sites associated with culture and history of the community can be considered heritage tourism. This view can be confirmed by the definition [14], that tourism is oriented to what we inherited, no matter whether these are historical buildings, craftsmanship or beautiful landscape. This interpretation of this term, however, was criticized by Y. Poria et al. [20], who claim that heritage tourism is about tourist motivation rather than attributes of a given place. No matter what the conclusion is in this discussion, many researchers support the definition proposed by P. Yale and treat heritage tourism as "nothing more" than tourism centered around what we inherited, which can mean anything, starting from historical buildings, through art to beautiful landscapes [21].

To sum up the discussion on heritage tourism and cultural tourism, one can conclude that they are closely linked. An example may be the use of heritage goods in shaping cultural tourism activities. Therefore, culture can be perceived as material and non-material goods, while heritage as this manifestation of culture, that should be maintained for future generations, as unchanged as possible.

A specific type of heritage are objects related to the industry being present in past centuries. Infrastructure now closed plants, production facilities, or extraction enterprises, are permanently inscribed in the infrastructure of numerous cities. In the past, these places aroused varied emotions. At first, they were associated with professional work and source of income for many families, a way to escape poverty. In turn, once unprofitable or unnecessary plants were closed, they were associated with negative emotions, due to work loss and growing unemployment. They are, however, a unique record in history, presenting technical and technological processes [22], and in this scope, they implement the cognitive need among tourists. Interest in the post-industrial heritage concerns both travelers (including tourists) as well as local communities the ancestors of which worked in the facilities being now tourist attractions. A crucial aspect of industrial heritage are customs and traditions shaped in the work environment of decades long gone, as well as approved behaviors and work ethics. In many cases they constitute vital elements of regional identity.

Tourism in industrial heritage areas is defined in many ways. In the global literature, one can find such terms as industrial tourism, post-industrial tourism, industrial heritage tourism, or industrial objects cultural tourism. Yet, the term industrial tourism [23] is the most popular. Although these terms are similar and often used interchangeably, researchers indicate slight differences between them. For example, M. Kronenberg [24] differentiates industrial and post-industrial tourism. According to him, the first one means tourism in active production plants that has educational and cognitive purposes, while the latter concerns traveling to places, where the industrial production was decommissioned. Yet, these terms are often treated as synonyms and mean tourist activity in areas where industrial heritage is the main theme of the travel. In addition, in this paper we understand industrial tourism similarly to A. Otgaar [25]—as visits to sites like museums, parks or other infrastructure, based on the active or abandoned industrial enterprises, which now fulfill a new function.

Apart from the mentioned theoretical references to industrial heritage tourism, in literature the empirical and theoretical context is discussed often. Y. Xu and Y. Cao [26] present experiences in the scope of economically justifiable use of industrial areas, as an example using, among others, German cases (in particular Ruhr region). F. Merciu et al. [27], in the publication on industrial heritage, indicated the necessity to maintain of local heritage and to maintain its economic sense. It is particularly important when it comes to city revitalization. S. Ćopić et al. [28] noticed that there are many industrial areas where tourism can be promoted as useful regional restructuring and economic development tools. As an example, Ruhr region (Germany) is where the major structural changes took place, due to the decline of extraction and the steel industry. The post-industrial areas were transformed into tourist attractions that can be found in the central part of Ruhr region in the area of Emscher park. They also noticed that it is a good example of sustainable growth, where tourism was used as a revitalization and industrial heritage protection tool. This positive approach and experience, as seen in Germany, can be applied to similar areas and regions in Europe. S. Vukosav et al. [29] first of all point to the fact that industrial heritage can have many functions and purposes, but its role in the general development of a community depends on the needs and priorities of its representatives. Moreover, they indicate that revitalization projects are based on inter-sectoral partnerships and their implementation requires engagement and cooperation of private sector and state support. According to the researchers, local authorities should play a key role in such processes, as these are the local authorities that assess which projects bring the best investors and what is of interest for the city.

From among the experiences of multiple post-industrial monument routes mentioned in the literature, it is evident that the protection of industrial heritage, and making it available by tourist enterprises is associated with increased tourist interest. The reasons for this are, originality, unique architecture, sentiment, or technical values. Making the heritage available for the public results in the emergence of tourist attraction. Yet, the process of transformation of a ruined factory, from the 19th century into a tourist attraction, is not easy and requires resources (especially capital) and the implementation of many managerial solutions based on well-thought business model.

## 3. Characteristics of Business Models

The business model concept, although widely described in literature [30,31], is understood intuitively or by way of using selected individual strategic tools among management practitioners. Similarly, in the tourism sector, formalized business models, based on clearly designed and well-thought elements, are rarely observed. However, as indicated by an analysis of many organizations, created by persons that have no educational background in terms of management, the lack of knowledge is not an obstacle when it comes to creating and implementing interesting business models. That is because almost every business starts with more or less a formal plan, drawn up on the basis of own ideas and experience of the venture initiator. However, what is being missed is the fact that the order in this scope constitutes added value, being the awareness of the significance of each step of business creation and the avoidance of mistakes and errors. It is, therefore, an important element for running a business that quite often makes it possible to survive on the market and to achieve a competitive advantage.

Modern business models take different forms when it comes to the links between components. They can be perceived as a synthetic description of the business [32] or as a tool [31], as a characteristic of relations between the components that lead to the development and capturing of value by the organization [33].

Informal business plans are being prepared by the humanity for thousands of years. Yet, only from the second half of 20th century, an attempt to name this process can be observed [34,35]. At first, the concept of a business model did not have a managerial meaning, but functioned in the context of business games. In the context of management, this concept appears in the literature as late as the mid-1970s [36], when E. Konczal [37] added a managerial value to business models, clearly suggesting that they should not be perceived only as scientific or natural science tools. Since 1970, business models

started being associated with business. This is confirmed by the fact that in the 1980s, the notion of dominant logic, the mental map of managers and a road map resulting from it, covering the logic of resources used to achieve business success had emerged [38].

In research literature on tourism, this concept rarely appears. In general, the particular components of business models of tourist enterprises are being indicated, that is innovation [39], relations with customers [40], creating value for the customer [41–44], or building cross-organizational networks [45].

An interesting discussion on business models in tourism sector concerns the accommodation and catering services. M. Diaconu and A. Dutu [46] paid attention to the evolution of hotel industry towards innovative business models, while N. Langviniene and I. DaunoraviÞinjtơ [47] in their publication listed a number of factors that need to be taken into account when creating a business model to be successful in the hotel industry.

A business model can be exemplified by the *smart tourism (ST)* concept. It is based on transforming huge amounts of data received via applications (most often mobile ones) into proposals of values for the customer [48]. The idea is about gathering knowledge on the preferences of consumers of tourism services, not only to transform it into new proposal of value, but also into a customized product, an efficient communication channel and a personally selected manner of relations buildings.

Currently, a business model is defined by D. Teece [31] as a tool describing the design or architecture of creation, supply, or value capturing mechanisms. The core of business model is defining the way in which the enterprise captures the value for the customers, entices them to pay for this value and converts payables into profits. S. Prendeville and N. Bocken [49] described it as a conceptual tool describing the activities that refer to business transactions between customers, partners, and suppliers, and the organization and their participation in the development and capturing of value. Slightly different approaches are presented by M. Geissdoerfer, P. Savaget and S. Evans [50]. According to them, business models are a simplified presentation of organizational elements (including interaction between them) defined to analyze, plan, and communicate in a complex organization structure.

An important reference to business models is the value triangle developed by R. Biloslavo, C. Bagnolii, D. Edgar [51], that covers the interactions between society, capital, and product, thereby creating three distinctive values—public value, partner value, and customer value. It is a proposal of value creation by sustainable business models, taking into account sustainable development to which more and more research on business models refers [52].

The most popular concept of a business model is CANVAS, created by A. Osterwalder and Y. Pigneur [30]. They indicate nine components describing a business model. These are market segments, proposed value for the customer, distribution channels, relationships with the customers, revenue streams, key resources, key activities, key partnership, and costs structure. Grouped components make it possible to visualize business model value and performance.

Yet, such static business models are criticized for their lack of experiments on, and modifications of, the components of business models, made by some entrepreneurs. Therefore, quite often dynamic business models are developed that are an alternative in the turbulent economic environment [53]. The proposal of dynamic business models is based on the combination of conventional schemes of business model [54] with modeling of system dynamics. By mapping key elements of value creation processes into the cause-and-effect relationships, with the use of simulation, it makes it possible for analysts of strategy, and entrepreneurs, to experiment and learn how the company reacts to strategic and organizational changes in terms of performance, innovation, and value creation.

## 4. Research Methodology

Based on the experience described in the literature and observed in industrial heritage tourist enterprises (IHTE), associated within Industrial Monuments Route (IMR), an identification of business creation models typology in post-industrial tourism, based on the criterion of the method of their creation was made. As a result, the subjective scope of the research covers post-industrial tourism

objects operating in Southern Poland, functioning within a formalized route administered by a regional government unity that plays the role of a coordinator. Currently, IMR is the biggest thematic route in Poland. It organizes 42 post-industrial tourist objects of highly diversified portfolio of the presented theme—starting from the Historic Coal Mine GUIDO, through the Museum of the Production of Matches, Radio Station and ending on breweries or adits. The listed sites are presented in Figure 1. At the same time, it is the only route in Central Europe that belongs to the European Route of Industrial Heritage (ERIH). In general, IMR was chosen because it is the most representative tourist route in Poland, which is based on industrial heritage with a well-organized structure.

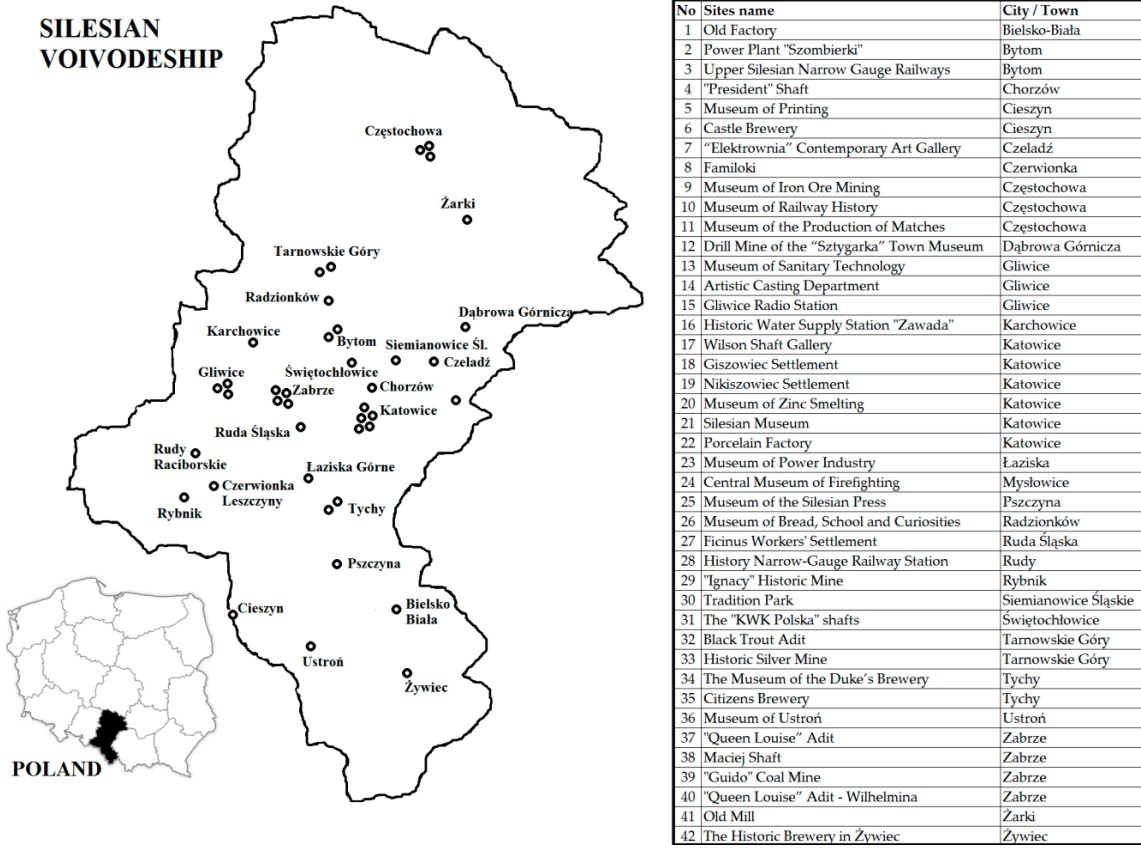

| No | Sites name | City / Town |
|---|---|---|
| 1 | Old Factory | Bielsko-Biała |
| 2 | Power Plant "Szombierki" | Bytom |
| 3 | Upper Silesian Narrow Gauge Railways | Bytom |
| 4 | "President" Shaft | Chorzów |
| 5 | Museum of Printing | Cieszyn |
| 6 | Castle Brewery | Cieszyn |
| 7 | "Elektrownia" Contemporary Art Gallery | Czeladź |
| 8 | Familoki | Czerwionka |
| 9 | Museum of Iron Ore Mining | Częstochowa |
| 10 | Museum of Railway History | Częstochowa |
| 11 | Museum of the Production of Matches | Częstochowa |
| 12 | Drill Mine of the "Sztygarka" Town Museum | Dąbrowa Górnicza |
| 13 | Museum of Sanitary Technology | Gliwice |
| 14 | Artistic Casting Department | Gliwice |
| 15 | Gliwice Radio Station | Gliwice |
| 16 | Historic Water Supply Station "Zawada" | Karchowice |
| 17 | Wilson Shaft Gallery | Katowice |
| 18 | Giszowiec Settlement | Katowice |
| 19 | Nikiszowiec Settlement | Katowice |
| 20 | Museum of Zinc Smelting | Katowice |
| 21 | Silesian Museum | Katowice |
| 22 | Porcelain Factory | Katowice |
| 23 | Museum of Power Industry | Łaziska |
| 24 | Central Museum of Firefighting | Mysłowice |
| 25 | Museum of the Silesian Press | Pszczyna |
| 26 | Museum of Bread, School and Curiosities | Radzionków |
| 27 | Ficinus Workers' Settlement | Ruda Śląska |
| 28 | History Narrow-Gauge Railway Station | Rudy |
| 29 | "Ignacy" Historic Mine | Rybnik |
| 30 | Tradition Park | Siemianowice Śląskie |
| 31 | The "KWK Polska" shafts | Świętochłowice |
| 32 | Black Trout Adit | Tarnowskie Góry |
| 33 | Historic Silver Mine | Tarnowskie Góry |
| 34 | The Museum of the Duke's Brewery | Tychy |
| 35 | Citizens Brewery | Tychy |
| 36 | Museum of Ustroń | Ustroń |
| 37 | "Queen Louise" Adit | Zabrze |
| 38 | Maciej Shaft | Zabrze |
| 39 | "Guido" Coal Mine | Zabrze |
| 40 | "Queen Louise" Adit - Wilhelmina | Zabrze |
| 41 | Old Mill | Żarki |
| 42 | The Historic Brewery in Żywiec | Żywiec |

**Figure 1.** The IMR sites on the Silesian voivodeship map. (Source: The Industrial Monument Route information [55]).

The research was carried out by way of face-to-face interviews and multiple case study of a representative character. The selection of this method was dictated by the possibility of using varied source materials, such as reports, offers, online resources, feedback received during interviews, and own observations. The research process was about preparing the literature and document query for selected IMR objects. The entities selection criteria were varied in terms of implemented business models. Then, based on the business model concept of A. Osterwalder and Y. Pigneur [30], several elements of the changing business model in post-industrial tourism enterprises were identified.

In general, the research process included four stages. The first stage was the analysis of the place of business. Two types of enterprises were selected. Some of the companies were just tourist-oriented, and the others were industrial enterprises, with an additional touristic function. The second stage was the analysis of the enterprise's history. We checked reasons for the sites being built and what function (production or not) they had. The third stage was to develop a typology of industrial tourism companies. During this stage multiple case studies and in-depth interviews were conducted. In the

last stage, a description of the identified types of enterprises and exemplary elements of components for business models were prepared.

## 5. Typology of Approaches to Business Models in Post-Industrial Tourism

A distinctive element of post-industrial heritage tourism objects, that distinguish such objects from other tourist attractions, is their focus on technical and industrial heritage. Based on this criterion, it was possible to divide the analyzed organizations into three types:

- post-production tourist organizations,
- production and tourist enterprises,
- tourist thematic organizations (The terms used towards organization forms (enterprise, entity or organization) are not incidental. In the case of production and tourist organizations, the production is implemented within the economic activity of the production company, therefore, we can identify an enterprise in this case. Yet, in case of the other two types (post-production and thematic organizations) the legal form can vary, that is why they are referred to as entities (implicitly economic, social, cultural, non-governmental, administrative ones etc.) or in general as organizations.).

The adopted division is also based on the characteristics of the place where the tourist activity is carried out and the intensity of tourist traffic presented figuratively. It can be thus specified that post-production organizations function within former (non-operating) production sites, production and tourist enterprises (combined) carry out both industry and tourist services, while thematic entities make use of industrial and technical heritage in a place usually not associated with the presented heritage.

The above types of business activity, being at the same time as the introduced business model, can be described from the context of a widely discussed concept of organization life cycle [54,56,57]. For that purpose, the concept of the life cycle of an organization was completed with an additional transitional stage that takes place once the basic production activity was finished or when it is taking place, that is at the same time the moment when the new activity is being initiated. When analyzing the historical changes in the studied objects, it became evident that its concerns both the domination and development of tourist function. This concept was completed to identify key stages of tourist industrial enterprises emergence basing on industrial heritage.

### 5.1. Post-Production Tourist Organizations (PPTO)

Post-production enterprise functions on the basis of past enterprises that operated in the industry. It is characterized by two stages of activity. The first one concerns the period when the production function dominates. The second stage is characterized by the domination of tourist function that is preceded by the development of implementation of tourist function. The post-production enterprise model is presented in Figure 2.

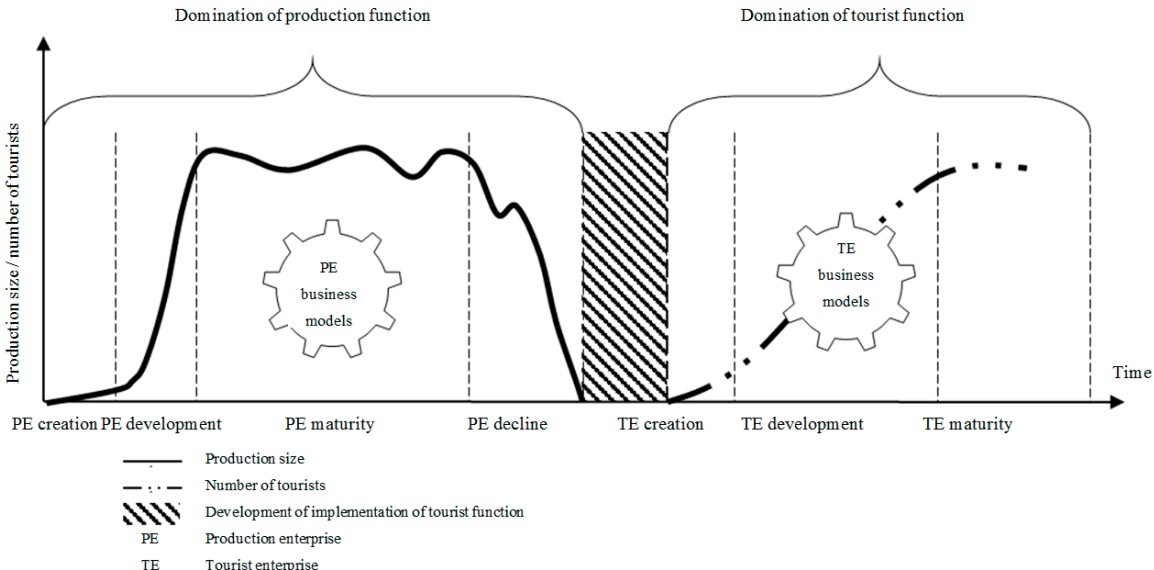

**Figure 2.** General model of post-production enterprise. (Source: Author's own work).

During the analyses of IMR it was noticed that during the first stage, the enterprise carries out a production or service function, generally understood as extraction, production of goods, and services. Decommissioning this activity results in the necessity to make a decision on the future of the enterprise being closed. Such an enterprise can be liquidated and its industrial infrastructure (or its part) can be removed or maintained. In case the infrastructure is to be maintained and made available for the tourists, the dominant function changes and a new business stage begins when the tourist activities become a crucial or strategic source of revenue.

An example of such an object is a very popular IMR tourist attraction that is a Historic Coal Mine, GUIDO, that finished extraction 89 years before the idea to transform it into a museum emerged in 2007. Mining levels at 170 and 320 m below the ground, leading to an authentic elevator, make it possible to discover how the work of miners looked like in 19th and 20th century [58]. Another example is the 19th century Black Trout Adit being a World Heritage Site and entered on the List of Historic Monuments.

One must pay attention, however, that in the presented stages two different business models are being used. The first one concerns production, the second one concerns tourist activities. Resources, key activities, as well as the server market segment are different just like the values proposed for the end customer. The stage before the second stage of enterprise functioning is defined as the *development of implementation of tourist function* is a time when a fundamental change of enterprise business model takes place. It is a time when the decisions on the scope of heritage are made available, the first concepts of object availability are presented, the funds to make the object available are sourced, the enterprise management is appointed (if the management ceased its activity), or the object is merged with an existing entity.

In post-production enterprises, the idea of implementing tourist function does not always take place once the production stopped. It can be postponed in time, when the production and tourist function can overlap (Figure 3).

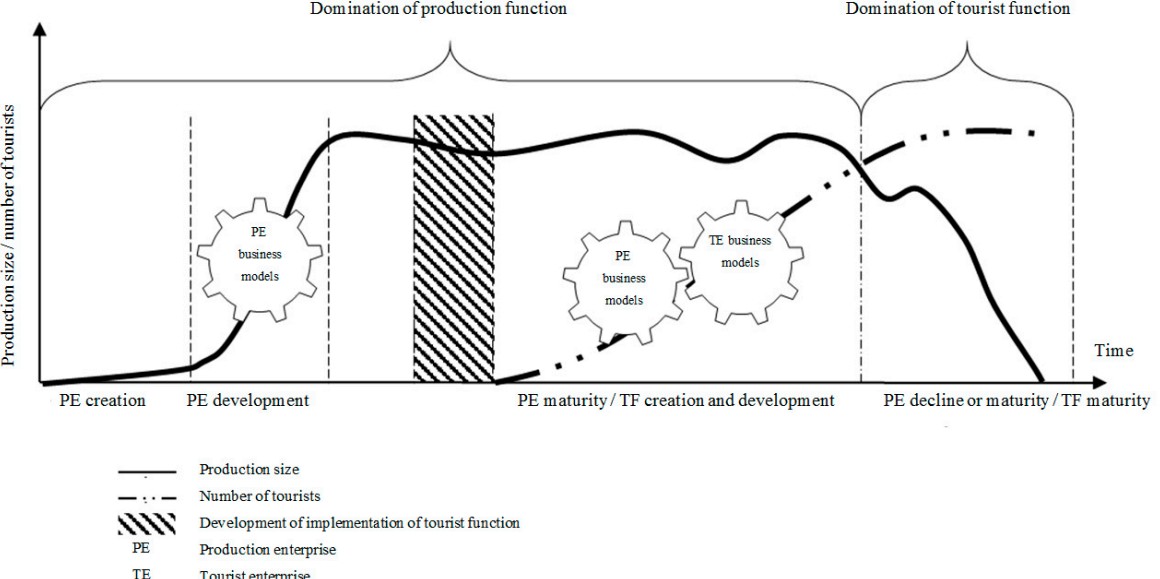

**Figure 3.** The postponement of tourist function in post-production enterprise model. (Source: Author's own work).

The postponement of the implementation of tourist function is characterized by the presence of two separate business models: Production and service (tourist), or a combination of these two activities in one integrated model. The aim of such enterprises is usually well-thought and well-maintained value proposal for the end recipient—customer and tourist.

Integrated business models, covering two types of activities, can be based on common components, for example, resources, customer segments, and treating the tourist activity as a product that completes the entity's offer.

However, post-industrial tourism enterprises do not only function on the basis of past industrial enterprises, where primary production was decommissioned or soon will be decommissioned. They can carry out tourist activity alongside the current industrial activity as a production and tourist enterprises.

## 5.2. Production and Tourist Enterprises (PTE)

Production and tourist enterprises carry out the tourist and production activity at the same time and both these activities are a source of revenue. Similarly, as in the case of post-production enterprises, the development of these economic entities is characterized by two stages of activity. In the first stage, the production function is the dominating one. Only later is it completed with tourist function. Yet, in general, there comes a moment when the two activities carried out at the same time are both strategic activities. The general model of a production and tourist enterprise is presented in Figure 4.

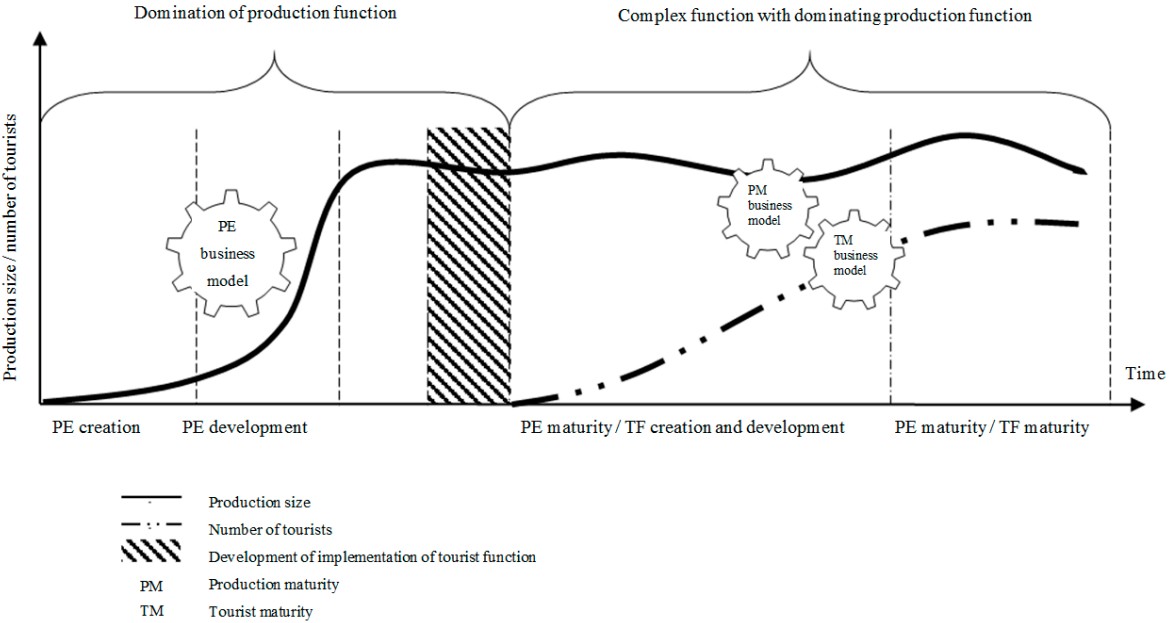

**Figure 4.** General model of a production and tourist enterprise. (Source: Author's own work).

The enterprises carrying out production and tourist activities at the same time are characterized by two business models interacting with one another: The Tourist activity model and production activity model or a complex business model that covers two separate areas of business activity. It will not, however, be an integrated model as particular components will be different for both activities. It is then possible to develop an autonomous model, for example, with shared resources or even a separate model that describes both activities separately. A sound solution for such a division may be also in implementing the concept of corporation separation, where the activity related to maintaining relations with customers, and the activities focused on product innovation development and infrastructure, are identified.

From the analysis of IMR objects, in the majority of cases of activity separation, the production model is dominant. The revenues on these activities are a strategic source of business financing. An example of such type of objects are breweries within IMR that carry out both beer production and offer sightseeing tours. For that purpose, The Museum of the Duke's Brewery in Tychy and Żywiec Brewery Museum were opened, where not only one can learn about the modern and historical process of beer brewing, but it is also possible to participate in workshops and presentations [59,60].

A particular example of such a model are enterprises where the transformation of the dominant activity is taking place. Then a slightly different second stage of activities can be observed. It can be observed by the decrease of production turnover and simultaneous growth of number of tourists (Figure 5). The decrease of production can result from various reasons, for example it can be due to plant restructuring or the change in the final product.

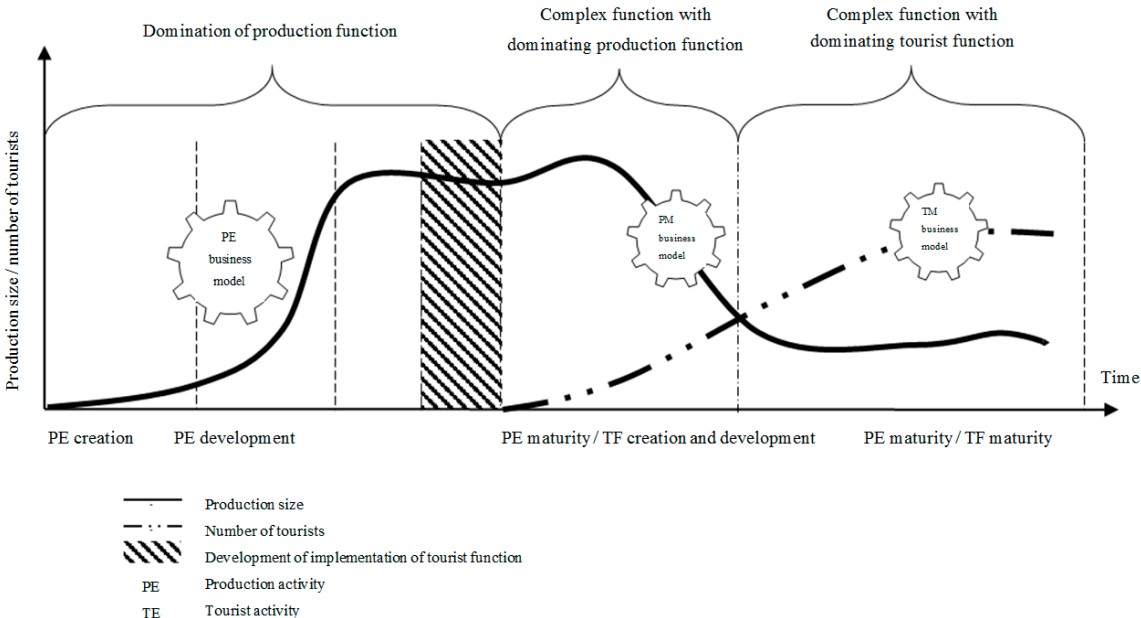

**Figure 5.** Production and tourist enterprise model with the dominant tourist function. (Source: Author's own work).

Another example of production and tourist enterprises are ones that, after a while since decommissioning the activity, re-started their original production activity. However, the restarted production is quite often limited in scope as it is not intended for production purposes, but has promotional or demonstrative character, and is about extending the value for the tourist (Gravari-Barbas, 2018). Unlike the previously described production and tourist enterprises, this activity is not characterized by two business models, only one (Figure 6).

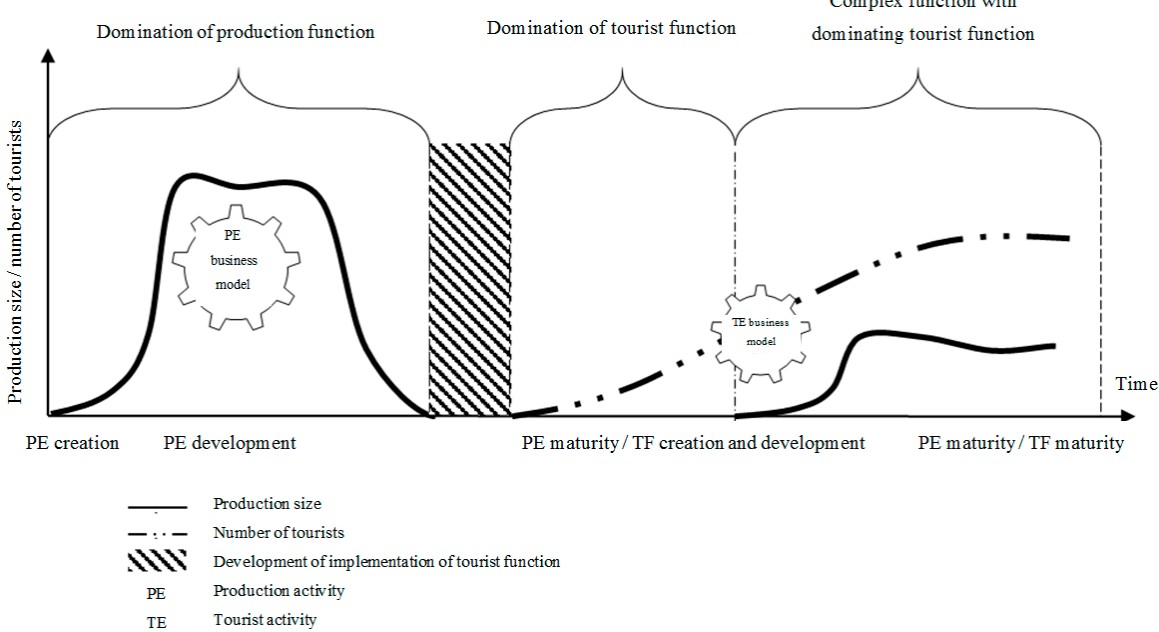

**Figure 6.** Production and tourist enterprise model where the production was re-started. (Source: Author's own work).

### 5.3. Tourist Thematic Organizations (TTO)

The last group of entities within IMR are enterprises, cultural, and art institutions providing services in the form of thematic exhibitions. First of all, they are characterized by activities devoted to industrial or technical heritage in places that in the past did not carry out any production (Figure 7).

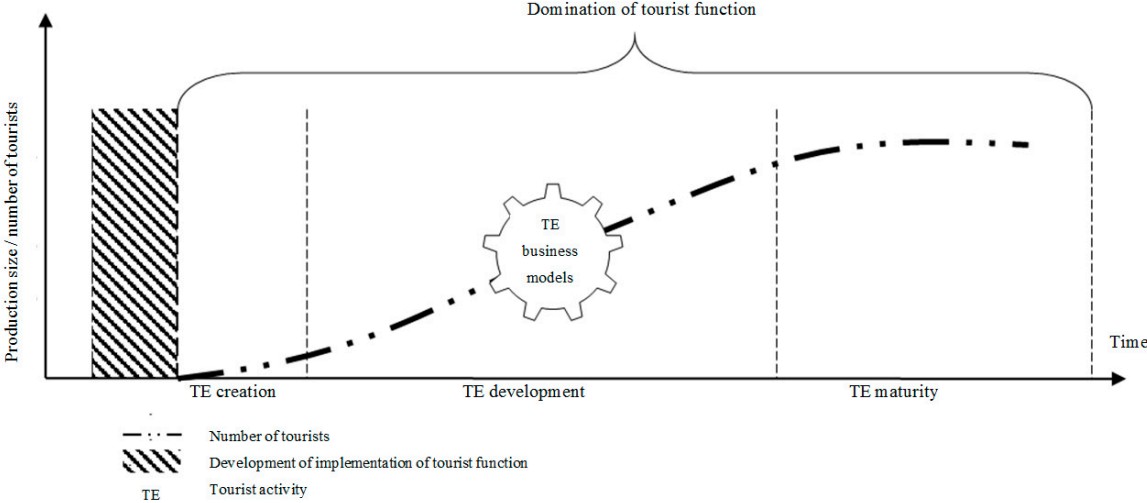

**Figure 7.** Thematic enterprise. (Source: Author's own work).

What is characteristic for this type of entity is only one dominant function—tourism. Such enterprises present the heritage on the basis of acquired resources, the subject of which is consistent with the assumed business profile. It can be assumed that these are objects that make the museum pieces available and exhibit tools, machinery, industrial, and craft devices from the past centuries.

There is no continuation or relation between the two functions of the activity being carried out. It is, therefore, a separate entity type of varied form of ownership and economic classification.

As an example, the Central Museum of Firefighting in Mysłowice [61], makes a historic collection on Polish firefighting available. Thus, visitors can learn more about the fire-fighters' work in the 19th and 20th century.

## 6. Discussion

The presented models of post-industrial heritage tourist organizations differ in many aspects, although several of them are characterized by significant similarities. It is worth analyzing the business model components and their relations in the context of the three mentioned schemes (Table 1).

In case of the infrastructure required in each of these models, one can list the historical devices, tools, and objects, including the tourist routes. Yet, almost in every model, the approach to these resources will be slightly different. In case of the production and tourist enterprises (PTE), the production model (PE) will complement the tourist model (TE), therefore the infrastructure of such enterprises will be larger and will require more engagement of activities and a wider circle of business partners. The situation is different in case of post-production tourist organizations (PPTO), as the role of resources is played not only by museum pieces, but also by the very post-production object, for example underground tourist routes in mining excavations. Preparation and maintenance of such a route requires many activities and engagement of significant group of partners (including local and regional administrative units). The smallest infrastructure will be present in thematic tourist organizations as the placement of exhibitions usually does not take place in historical buildings, but in specially prepared rooms, that make it possible to discover the thematic path.

**Table 1.** Exemplary elements of components for business models.

| Components | Post-Production Tourist Organization (PPTO) | Production and Tourist Enterprise (PTE) | | Tourist Thematic Organizations (TTO) |
|---|---|---|---|---|
| | | *Production (PE)* | *Tourism (TE)* | |
| | | *Infrastructure* | | |
| **Key Activities** | Activities aiming to adjust the routes and exhibitions to the traffic route and maintain a high quality of service | Production activities | Separating tourist routes from production lines Securing mutual interaction of both activities | Acquiring museum pieces and creating thematic routes |
| **Key Resources** | Historic machinery and industrial devices, craft tools Routes in disused excavations Tradition and history of the plant | Production and transport base | Machinery, showpieces and routes Tradition and history of the industry | Showpieces, traditions, history |
| **Partner Network** | Guides and retired employees of the plant Hotel and catering industry Regional administrative unit | Suppliers Distributors Sellers Service | Guides Hotel and catering industry Regional administrative unit | Guides Hotel and catering industry Regional administrative unit |
| | | *Offering* | | |
| **Value Propositions** | Learning about the environment where the ancestors worked Learning about the work ethos of the region Learning about the industry history Cultural experiences | Value resulting from the usability of products | Learning about the current work environment Learning about the past and present production process | Familiarizing oneself with the subject of sightseeing Cultural experiences |
| | | *Customers* | | |
| **Customer Segments** | Tourists visiting this region Residents Educational institutions | Recipients (retail and wholesale) of production | Tourists and residents Educational institutions | Tourists and residents Educational institutions |
| **Channels** | Internet, local press | Internet, direct sale, advertising in media | Internet, direct sale, advertising in media | Internet, direct sale |
| **Customer Relationships** | Tourist—"guest" Making new attractions available gradually and periodically Creating new sightseeing programmes | Improving the quality of goods and their distribution Establishing the brand and image | Tourist—"guest and potential customer" Brand creation | Tourist—"guest" Telling the story of the exhibition |
| | | *Finances* | | |
| **Cost Structure** | Cost of post-production infrastructure and its adjustment Showpieces maintenance | Cost of production and sales chain | Cost of tourist route maintenance Cost of production safety Cost of promotion and servicing | Cost of thematic route maintenance Showpieces servicing |
| **Revenue Streams** | Revenues on tourist and cultural activity Subsidies | Sales revenues | Revenues on tourist and cultural activity Subsidies | Revenues on tourist and cultural activity Subsidies |

Source: Author's own work.

Differences can also be seen in the value for the customer. Apart from the value of the product that is offered by tourist and production enterprises, repeatability of cognitive value of the tourist routes can be seen. The dominating value is the one related to the knowledge about history, personal past, and regional culture. Additionally, in some cases, cultural experiences in a wider scope are possible. It results from the fact that both post-production and thematic organizations include cultural events (concerts, meetings with popular people, various events etc.) in their offer.

The characteristics of the customer in all models associated with tourism will be directed towards the tourist as a guest whose cognitive needs become the key task for the team servicing the tourist traffic. Nevertheless, as in the case of production and tourist enterprises, the role of the sightseeing route will be more significant as it serves not only cognitive values but also promotional ones. The knowledge

about the entity offering a given product and about the production process of this good can attach the visitor not only to the product but also to the manufacturer. It can result in loyalty to products that are available every day. Extremely crucial is also the development of the relationship with the customer in other types of industrial heritage tourist organizations, for example, by developing routes with the theme of history.

What can also be noticed is the different financial structure in the discussed organizations. These differences stem not only from the levels of the already mentioned costs but also from revenues as in the case of entities not supporting themselves from the sale of their own products, they can obtain subsidies for cultural activities as the only source of revenue. State support is also necessary when restoring the usability of an object that has been closed for many years.

It is worth noting, that in the case of production and tourist enterprises, it is also necessary to take into account the costs of business and tourist safety. It results from the necessity to maintain production continuity despite the tourist traffic as well as from the need to protect the tourist in the vicinity of machinery and devices and in the excavations.

All discussed business models seem to have their strengths and weaknesses. For example, managers of PPTO enterprises indicated that they can show heritage only by exhibits, without real production. On the other hand, PTE managers indicated that showing production lines requires increased caution in order to ensure the safety of tourists. However, managers have limited influence on what type of enterprise their business will become. The transformation from one type of enterprise to another is impossible in the short-term period, in which business models are usually considered. Therefore, managers should focus on improving business models rather than changing them to another one.

## 7. Conclusions

The article discusses three basic types of business transformations identified in post-industrial heritage tourist entities. The post-production organization model can be considered the most popular scheme on the analyzed route. It concerns an enterprise or cultural institution that previously was a production or extraction plant and currently services tourists. Although these objects were not designed with tourists in mind, they perfectly fulfill this function thanks to the presented transformations.

Thanks to this, the care of post-industrial heritage becomes an interesting implementation of the principles of sustainable development. Activities aimed at creating tourist objects of cultural heritage, can save the legacy of previous generations from oblivion, and at the same time preserve them for future generations.

This research opens the way for further exploration of industrial heritage tourism in Poland. It can be useful for managers, especially when one compares our results with well-researched examples from Germany, France, or Belgium. We believe that the results of these comparisons will bring many conclusions to the discussion on creating customer value based on industrial tourism.

**Author Contributions:** A.R.S. developed the methodology, introduction and conclusions, cited examples of post-industrial heritage tourist enterprises and compared the identified post-industrial enterprises transformation model types, as well prepared the final contents of the article and proofread and revised it. K.H. reviewed the literature in terms of business models and proposed a division of post-industrial enterprises transformation models as well as prepared the characteristics of the Industrial Monuments Route in Poland.

**Funding:** This research received funding by the National Science Center in Poland.

**Acknowledgments:** This paper was published as part of the research project 'A business model for health resort enterprises' No. 2017/25/B/HS4/00301, supervised and financed by the National Science Center in Poland and as part of statutory research ROZ 1:BK-231/ROZ1/2018 (13/010/BK_18/0029) at the Silesian University of Technology, Faculty of Organization and Management.

**Conflicts of Interest:** The authors declare no conflict of interest. The funders had no role in the design of the study; in the collection, analyses, or interpretation of data; in the writing of the manuscript, or in the decision to publish the results.

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
