# Peer review of "A Business Creation in Post-Industrial Tourism Objects: Case of the Industrial Monuments Route"

_sustainability, doi:10.3390/su11051451_

Round 1

Reviewer 1 Report

This is a really interesting paper, which can attract attention of the Sustainability's readership. I really like the model typology suggested by the authors – this seems to be a great contribution to the available knowledge. I recommend to accept it after certain improvements. My recommendations are specified below.

1)      Please, make the title shorter and more appealing.

2)      Please, avoid numbering and formal structuring of the abstract.

3)      The authors should explain sharply that they prefer very narrow definition of industrial tourism (as based on the only active enterprises) and oppose it to post-industrial tourism (as based on the only closed enterprises). However, the broad and common definition of industrial tourism covers the both issues, i.e., the active and abandoned enterprises.

4)      Too little literature sources are cited in Introduction. The authors should include the basic works devoted to industrial tourism and heritage, including these ones: Otgaar,  A. (2012). Towards a common agenda for the development of industrial tourism. Tourism Management Perspectives, 4, 86-91. Jonsen-Verbeke, M. (1999). Industrial heritage: A nexus for sustainable tourism development. Tourism Geographies, 1, 70-85. Yashalova, N.N., Akimova, M.A., Ruban, D.A., Boiko, S.V., Usova, A.V., & Mustafaeva, E.R. (2017). Prospects for Regional Development of Industrial Tourism in View of the Analysis of the Main Economic Indicators of Russian Tourism Industry. Economic and social changes: facts, trends, forecast, 10, 195-213. Ruban, D.A., Tiess, G., Sallam, E.S., Ponedelnik, A.A. & Yashalova, N.N. 2018. Combined mineral and geoheritage resources related to kaolin, phosphate, and cement production in Egypt: Conceptualization, assessment, and policy implications. Sustainable Environment Research. 28: 454-461.

5)      The first sentence of the section 2 should be accompanied by citations. Please, check also the other places.

6)      The section 6 MUST be named Discussion.

7)      Why not to split Discussion into two sub-sections?! The first would bear the already-available comparison, and the second would contain the comparison of the outcomes of the present study with the results of the other, somewhat similar studies in the other countries and regions. E.g., industrial heritage is well-exploited in the Ruhr region of Germany – why not to consider, especially because mentioned earlier in this paper?

8)      Conclusions should be shortened and made more concise. Please, avoid too lengthy and repetitive explanations, and better list the only main findings and perspectives for further research.

9)      I suggest to search for the literature on sustainable organizational development and to cite it in this article relatively to the considered models (this may become the third sub0section of Discussion). As the word 'sustainable' appears in the title, it is sensible to present the relevant discussions in the text. E.g., see this one: Lawler E.E. III, Worley C.G. Designing organizations for sustainable effectiveness // Organizational Dynamics. 2012. Vol. 41. P. 265-270.

10)  Please, do not forget to style citations and references strictly according to the journal's rules.

Author Response

Thank you for the review of the article.

In the article we focus on ‘business model’ more than ‘sustainability, so we corrected the title. It is shorter and more precise. Now it does not contain the word 'sustainable', which was confusing in the context of the conducted research. We removed the formal structure in the abstract. We also changed the name of section 6. In addition, the conclusions have been shortened by the repetition of the content of the article. Moreover, we added citation and the new reference and few words about future research.

You suggested to compare our research with examples from other countries. We agree that this could be good expansion to research. However, we did not want to expand the work with comparative analysis in order not to lose the uniqueness of the route. We will include this comparison in next article.

Best regards

Adam Szromek

Krzysztof Herman

Reviewer 2 Report

Dear Authors

The presented article carries out a study on the possible use of industrial facilities for tourist purposes in Poland. I find the proposed theme interesting and I believe that the approach of the article is very appropriate.

However, there are several issues that from my point of view should be enhanced and corrected in order to improve the article and make it suitable to publish it in Sustainability.

1.- I am not clear about the approach regarding the sustainability of the article. This makes me having doubts about the suitability of it for publication in Sustainability. It may be necessary to add in the analysis or in the conclusions a few paragraphs explaining how this analysis could help industrial heritage managers to ensure its sustainability and tourism use.

2.- In general, and continuing with the previous point, I am not clear about the objective of the article. The aim is to carry out an analysis of the possible business models that make possible to take benefit of industrial heritage for tourism purposes. But what is the point of this analysis? If it is merely descriptive, I see it as something very limited. But if its function is, as I said before, helping managers or companies, it should be explained. I suggest the objective to be explained in the final part of the introduction.

3.- There is a lack of a more elaborate and well-defined methodology. Section 4, in my opinion, is scarce. The IMR is explained, but it does not explain correctly how the research has been carried out. What have been the steps taken? What are the criteria for selecting the study sites? What are the chosen cases? In my opinion, this section should be undoubtedly and substantially improved.

4.- Section 5, on the other hand, seems to me to be very well developed. Very complete in general.

5.- However, section 6 is too long on a descriptive part (which does not necessarily have to be something negative) but it lacks a part of analysis. Is there a business model that is better than another? What are the advantages and disadvantages of each of them? Personally, I would also add in this section some recommendations and suggestions for the use of other possible industrial patrimonies and their reconversion into tourist attractions.

Some other suggestions:

- Review the use of capital letters in the title.

- Adapt the references to the format required by the magazine. If I am not wrong, the use of [number] format is recommended.

- Lines 53-55: please, provide some examples and references.

- Line 61: please, provide reference.

- Lines 69-70: please provide reference.

- Line 92: “the objects that present that heritage” sound strange to me. The same with line 93 “more and more”. Please, consider rewriting. Same with line 104 “history and history”. Although it can be understood, I suggest rewriting.

- Lines 168-169: please provide references (“widely described in literature”, where?) and consider rewriting.

- Lines 171-172: “as indicated by an analysis of many organizations”, which ones? References.

- Lines 226-230: do you refer to “Business Canvas model”? Anyway, it should be included in the section.

- Line 235: please provide reference.

- Section 4: I suggest starting the section with a better definition of what IMR is. Please, consider including a map and a table where the 42 post-industrial tourist attraction are lists. I suggest using this same table to mark those selected and included in the analysis. Please also explain how the selected sites where chosen and under which criteria.

- Lines 263-271: please, consider explaining this better.

- Lines 325-327: please, provide some examples from the selected cases.

Author Response

Thank you for the review of the article.

We corrected the title. It is shorter and more precise. Now it does not contain the word 'sustainable', which was confusing in the context of the conducted research. In introduction we explained how the article can help managers to understand how the value proposition can be created using specific resources and goal of this article was formulated more precisely.

In the methodology we added the characteristics of the research process. It includes stages and a description of steps that needed to be taken in order to finalize each stage.

Moreover, there were two reasons why we have choosen the IMR as an example. The first reason was that it is the only one route in Central Europe that belongs to European Route of Industrial Heritage (ERIH), second reason is that IMR is well-organized industrial touristic route in Poland and represents the highest managements level.

In our opinion the process of transformation industrial plants into tourist attractions is a separate topic, that requires further research, utilizing additional methodology. We intend to tackle this issue in our next article, for we believe that this topic is to vast to fit into one paper with the research we already described, and it requires to be handled separately. 

Among other we reviewed the article, corrected it and applied yours suggestions.

Best regards

Adam Szromek

Krzysztof Herman

Round 2

Reviewer 1 Report

I'm fully satisfied with the authors' improvements and tend to recommend acceptance. I just ask to present text citations strictly according to the journal's rules.

Author Response

Dear Sirs

Thank you for the review of the article.

We adapted the references to the format required by the magazine. Moreover, in section 4 we added figure with IRM sites on the Silesian voivodeship map.

Best regards

Adam Szromek

Krzysztof Herman

Reviewer 2 Report

Dear authors.

Congratulations on the new version of your article, which has undoubtedly gained in quality. I appreciate the effort made to incorporate the requested changes and respond to our suggestions. The resulting article may be accepted for publication in Sustainability.

However, I would like to add a few extra comments that I think could help transform this article into a better one, increasing its potential impact and interest to readers.

- Adapt the references to the format required by the magazine. If I am not wrong, the use of [number] format is recommended.

- Section 4: Please, consider including a map and a table where the 42 post-industrial tourist attraction are lists. I suggest using this same table to mark those selected and included in the analysis.

Author Response

Dear Sirs

Thank you for the review of the article.

In section 4 we added figure with IRM sites on the Silesian voivodeship map. Moreover, we adapted the references to the format required by the magazine.

Best regards

Adam Szromek

Krzysztof Herman
